# Trends of tuberculosis case detection, mortality and co-infection with HIV in Ghana: A retrospective cohort study

Eric Osei[1,2]*, Samuel Oppong[3], Joyce Der[3]

1 Department of Population and Behavioural Sciences, School of Public Health, University of Health and Allied Sciences, Ho, Ghana, 2 Department of Public Health Graduate School, Yonsei University, Seoul, Republic of Korea, 3 Department of Epidemiology and Biostatistics, School of Public Health, University of Health and Allied Sciences, Ho, Ghana

* eosei@uhas.edu.gh

## Abstract

### Background

In an era of renewed commitment to accelerate the declines in Tuberculosis (TB) incidence and mortality, there is the need for National Tuberculosis Programmes (NTPs) to monitor trends in key indicators across a geographical location and to provide reliable data for direct measurement of TB incidence and mortality. In this context, we explored the trends of TB case detection, mortality and HIV co-infection, and examined the predictors of TB deaths in Ten districts of the Volta region of Ghana.

### Methods

We conducted a retrospective cohort study of all TB cases registered from 2013 to 2017 in 10 districts of the Volta Region of Ghana. Case detection rate (CDR) was computed as the ratio of the number of new and relapse TB case notified to NTP to the number of estimated incident TB cases in a given year. Case fatality rates were estimated using data from 2012–2016 cohort of TB patients. Simple and multiple logistic regression were used to identify predictors of TB deaths with odds ratios and 95% confidence intervals estimated.

### Results

Overall, there were 3,735 new and relapse TB patients who commenced anti-TB treatment during the period, representing the case detection rate of 40.1% with district variations. The CDR remained stable during the 5 years. Of the total cases, HIV status was documented for 3,144 (84.2%), among whom, 712 (22.6%) were HIV positive. The TB/HIV co-infection was more prevalent among children under 15 years of age (30.1%), males (30.6%), treatment after lost to follow-up patients (33.3%), and smear-negative pulmonary TB patients (29.1%). The prevalence of TB/HIV co-infection did not significantly change over the years. The overall case fatality rate was 13% (n = 486), with considerable variation among HIV-positives and HIV-negative TB patients (21.8% and 11% respectively) (p<0.001) and among districts.

**Data Availability Statement:** All relevant data are within the paper and its supporting information files

**Funding:** The authors received no specific funding for this work

**Competing interests:** The authors have declared that no competing interests exist.

**Abbreviations:** AIDS, Acquired Immunodeficiency Syndrome; ART, Anti-retroviral Therapy; BMU, Basic Management Unit; CDR, Case Detection Rate; CPT, Cotrimoxazole prophylaxis Therapy; DOTS, Directly Observed Treatment Short-course; HIV, Human Immunodeficiency Virus; NTBCP, National Tuberculosis Control Programme; PLHIV, People Living with HIV; TB, Tuberculosis; WHO, World Health Organization.

TB/HIV co-infection, sputum smear-negative pulmonary TB and district of anti-TB treatment predicted TB mortality.

## Conclusion

TB case detection rate was low and remained stable during the study period, whereas co-infection with HIV and mortality rates were quite high, indicating the need for feasible strategies such as active case finding to improve case detection, and improved case management to reduce mortality.

## Background

Tuberculosis (TB) has burdened humanity for years and continue to pose a huge threat to public health and health systems [1]. Worldwide, about 10 million incident cases of TB occur in 2017. However, more than one-third of these cases were missed by the health systems and remain undetected [2]. Ensuring early diagnosis and prompt initiation of effective chemotherapy are key aspects of TB control programmes. Any delay in diagnosis and subsequent treatment of effective TB patients not only pose a high risk to communities but can also lead to more advanced disease state, which may result in more complications and death [3]. Recognising this, the Directly Observed Treatment Short-course (DOTS) strategy, introduced by the WHO, sets targets for all National TB Control Programmes to detect at least 70% of estimated infectious cases and successfully treat 85% of them in order to interrupt the transmission, reduce mortality and prevent emergence of drug resistance [4]. The End TB Strategy, which set target to reduce TB deaths by 95% and a 90% reduction in TB incidence by 2035, from the 2015 levels [5], also underscores the importance of early diagnosis, including universal drug susceptibility testing (DST) and systematic screening of contacts and high-risk groups to reduce the case-detection gap [6]. The strategy also set recommended targets level of ≥90% TB treatment coverage for all nations by 2025 at the least [7].

TB/HIV co-infection poses widespread diagnostic, management and economic challenges to many, particularly African countries, where the burden of HIV-associated TB is highest [7]. HIV infection has been known to increase the risk of developing active TB by accelerating disease progression or reactivating latent infection [8,9]. Additionally, TB/HIV co-infected patients, especially in the absence of antiretroviral therapy (ART), have a significantly worse prognosis. Knowing this, the WHO recommends systematic screening for TB symptoms among People Living with HIV (PLHIV) and vice versa, as an essential part of the care package, together with linkage to diagnostic and treatment services, where necessary [7]. In the WHO African region, about 87% of TB patients were tested for HIV in 2018, among whom, 29% were HIV-positive, representing 71% of the worlds TB/HIV co-infection burden [7].

Among all causes of deaths worldwide, TB ranked tenth, and it is the single most important cause of death from a single infectious agent since 2007. About 85% of TB deaths occurred in the WHO South-East Asia and African regions. In 2018, there were 15 HIV-negative TB mortalities and 3.3 HIV-positive TB mortalities per 100,000 population globally, and 37 and 20 per 100,000 population respectively in the WHO African Region [7]. The mortality rate of TB has been falling globally since 2000. Despite this feat, however, the world is not on track to achieve a 35% reduction in the total number of TB deaths by 2020, from the 2015 level, as set by the End TB Strategy. Reaching this milestone will require a faster rate of decline and case fatality rate not more than 10% by 2020 [7].

In Ghana, even though remarkable achievement has been made since adopting the DOTS strategy in the 1990s, TB remains a public health problem with a huge economic and health impact on individuals as well as the health system in general [10]. In 2017, there were an estimated 44,000 incident cases of TB in the country, however, only 14,550 (33%) cases were detected and notified to WHO, among whom, 21% were HIV-positive [2].

In an era of renewed ambition to accelerate the declines in TB incidence and mortality, there is the need for National Tuberculosis Programmes (NTPs) to monitor trends of key indicators across a geographical location, in order to provide reliable data for direct measurement of TB incidence and mortality [2]. This allows NTPs to report on the progress in reaching its goals and objectives. To the best of our literature search, no study in Ghana has examined the trends of TB case detection and mortality at the district level, as well as predictors of TB deaths. In this context, the present study explored the linear trends of TB case detection, mortality and co-infection with HIV, and examined the predictors of TB deaths across Ten districts of the Volta region of Ghana.

## Materials and methods

### Study settings and design

We conducted a retrospective cohort study of all TB cases registered from 2013 to 2017 in Ten TB Basic Management Units (BMU)/ districts of the Volta Region of Ghana. A BMU is defined in terms of management, supervision, and monitoring responsibilities. Each unit had one or more treatment centres, laboratories, and a hospital. There is a coordinator who oversees TB control activities for the unit and maintains a master register of all TB patients being treated. The Volta Region is one of the 10 administrative regions of Ghana located at the Eastern part of the country. The region is bound to the north by the Northern Region, the south by the Gulf of Guinea, west by the Volta Lake and the east by the Republic of Togo. It is divided into three natural geographical belts namely the southern, middle and the northern belts and has 25 administrative districts, 377 health facilities, serving a population of over 2,789,211. There are 44 DOTS centres and 41 laboratories in the region that provide TB diagnosis and treatment services to TB clients. At the DOTS centres, TB diagnosis, treatment, and monitoring are done as per the National Tuberculosis Control Programme (NTBCP) guidelines. As in all other parts of Ghana, there are also TB/HIV collaborative activities at all DOTS centres in the region which aim to reduce the burden of TB among people living with HIV (PLHIV) and vice versa.

### Sampling

For representativeness, stratified sampling approach was used to select participating districts. Strata were the geographical belts (southern, middle and northern) of the region. Five districts in the region did not have data from 2013, hence were excluded in the strata for selection. Ten (50%) districts were then selected from the remaining twenty. We randomly selected four districts each from the South (Keta, Ketu South, Central Tongu, and South Tongu), Northern (Nkwanta South, Krachi West, Kadjebi, and Krachi East); and two from the middle belt (Kpando and Hohoe) using probability proportion to the number of districts in each stratum. The data used to estimate CDR were that of patients with TB who were registered and commenced anti-TB treatment between 1st January 2013 and 31st December 2017 in the participating districts. Patients who were treated for TB between 2012 and 2016 constituted the population for estimation of mortality rate and its predictors.

## Data source and extraction

Data were extracted from the master registers of each BMU in April 2018. The registers are provided by the Ghana NTBCP and are used by all districts in Ghana. They are used to record data of all patients with TB being treated at all DOTS centres in the districts, which is used to monitor the programme and report on indicators to the higher level. Data contained in the register include patients' demographics; type and category of TB; initial and follow up sputum smear examination results; treatment outcomes; and HIV status and anti-retroviral treatment. Data were extracted directly into Microsoft Excel spreadsheet designed to capture all relevant variables. Data extractions were done by trained undergraduate students, supported by TB coordinators of participating districts. Variables such as age, sex, types of TB, co-morbidity with HIV/AIDS, and treatment outcomes were extracted for analysis. To ensure data quality and accuracy, the district TB focal persons were made to systematically check and compare each year's data extracted with the source document. Discrepancies were checked and corrected accordingly. Data were cleaned to check for consistency and completeness of the data set before exporting for analysis.

## Data handling and analysis

Extracted data were exported into Stata Version 13.1 (Stata Corp, College Station, Texas, USA) for analysis. Categorical variables were expressed in percentages and median and interquartile range (IQR) for continuous variables, they were compared using Pearson's chi-square test or Fisher's exact test and the independent-samples t-test accordingly to describe the study population. Chi-square for trend was used to examine linear trends. Simple and multiple logistic regression model with stepwise variable selection procedure were employed to identify independent predictors of TB deaths. Variables with p-value of <0.05 in the simple regression analysis were considered as candidate variables for inclusion in the multiple logistic regression analysis. The degree of association between dependent and independent variables was assessed using odds ratios (OR) with 95% confidence intervals (CI).

**Outcome measures and definitions.** Case Detection Rate (CDR) was computed as the number of TB cases notified during the period divided by the estimated annual TB incidence per 100,000 Ghanaian population [11] and was expressed in percentage. TB/HIV co-infection rate was measured as the percentage of TB patients tested for HIV who were HIV positive. Case fatality rate was assessed using data from 2012–2016 treatment cohorts of all TB patients. We used **the** WHO's standard criteria [12] to define TB patients as shown in Table 1.

## Ethical statement

Ethical approval was obtained from the University of Health and Allied Sciences Ethical Review Committee. Informed consent was not obtained since there was no direct contact with patients. Patients' identifying information were not collected from the registers. Permission was sought from the Volta regional health director and district directors of participating districts before data were extracted. Data extractions were done in the presence of district health staff.

## Results

### General characteristics of study subjects

There were a total of 3,735 TB cases of all forms enrolled on treatment during the period in all 10 study districts, of whom ages ranged between 1 and 96 years, with median (interquartile range) age of 44 (19–69) years. Children below the age of 15 years constituted 4.3% (n = 161)

**Table 1. Case definitions of tuberculosis.**

| Category | Definition |
|---|---|
| **Type of TB** | |
| Pulmonary smear-positive TB | Patient with at least two sputum specimens with sputum positive for acid-fast bacilli (AFB) by microscopy, or a patient with only one sputum specimen with smear-positive for AFB by microscopy and chest radiographic abnormalities consistent with active pulmonary TB. |
| Pulmonary smear-negative TB | Patient with symptoms suggestive of TB with at least two sputum specimens which were negative for AFB by microscopy, and with chest radiographic abnormalities consistent with active PTB or a patient with two sets of at least two sputum specimens taken at least two weeks apart, and which were negative for AFB by microscopy, and radiographic abnormalities consistent with pulmonary TB and lack of clinical response to one week of broad-spectrum antibiotic therapy. |
| Extra-pulmonary tuberculosis (EPTB) | TB of organs other than the lungs, such as lymph nodes, abdomen, genitourinary tract, skin, joints, bones, meninges, etc |
| **Patient category** | |
| New case | A patient who has never had treatment for TB before or has been on anti-TB treatment for less than four weeks |
| Relapse | A patient who has been declared cured or treatment completed of any form of TB in the past but who reports back to the health service and is found to be acid-fast bacilli smear-positive or culture-positive |
| Treatment failure | A patient who, while on treatment remained smear-positive or become again smear-positive at the end of the five months or later after commencing treatment. |
| Transferred in | A patient who started treatment in one health facility and transferred to the hospital to continue treatment and follow up |
| Treatment after loss to follow-up | Patients who have previously been treated for TB and were declared lost to follow-up at the end of their most recent course of treatment. (These were previously known as treatment after default patients.) |
| Others previously treated | Patients who have previously been treated for TB but whose outcome after their most recent course of treatment is unknown or undocumented |
| **Treatment outcome** | |
| Died | Patients who died from TB during treatment |

of all reported cases, while 15.4% (n = 574) were elderly (64+ years), among whom 63.6% were males and the rest females. Overall, males accounted for 62.5% (2,335) of all cases and the proportion of male cases increased with age; most of them (3,578; 95.8%) were new; 55 (1.5%) relapsed; 33 (0.8%) treatment after lost to follow-up; and 25 (0.7%) were treatment failure patients. Of all reported cases, 1,712 (45.8%) were pulmonary positive, 1900 (50.9%) were pulmonary negative, while 116 (3.1%) were extrapulmonary TB patients. With regards to HIV status, 712 (19.1%) of them were co-infected with HIV, while the HIV status of 591 (15.8%) was unknown. The majority (1125; 30.1%) were reported in the Ketu South district, followed by Keta (440; 11.8%), and the least (189; 5.1%) in Krachi East district (Table 2).

## TB case detection rate and trends

The overall case detection rate (CDR) of all forms of TB during the 5-years study period was 40.1%, (95%CI: 39.4–41.4), of which Kadjebi, Kpando and Ketu South districts were above and the rest below. Kpando (76.1%; 95%CI: 72.1–79.7) and Ketu South (77.4%; 95%CI: 75.2–79.5) had the highest CDR, while Krachi East (20.2; 95%CI: 17.8–22.9) and Krachi West (20.7%; 95%CI: 17.6–24.1) recorded the lowest CDR in the range of 20%, as shown in Fig 1. Case detection rate over the years remained fairly constant, with the highest rate of 42% recorded in the years 2013 and 2017, while 2016 had the lowest rate of 37% (Fig 2).

**Table 2. Background characteristics of TB patients (2013–2017).**

| Variable | Male (n = 2335) n(%) | Female (n = 1400) n(%) | Total (N = 3735) n(%) | $X^2$ p-value |
|---|---|---|---|---|
| **Age group(years)** | | | | |
| ≤14 | 80 (49.7) | 81 (50.3) | 161 (4.3) | <0.001 |
| 15–24 | 152 (48.9) | 159 (51.1) | 311 (8.3) | |
| 25–34 | 339 (56.4) | 262 (43.6) | 601 (16.1) | |
| 35–44 | 555 (65.1) | 297 (34.8) | 852 (22.8) | |
| 45–54 | 501 (67.4) | 242 (32.6) | 743 (19.9) | |
| 55–64 | 343 (69.6) | 150 (30.4) | 493 (13.2) | |
| >64 | 365 (63.6) | 209 (36.4) | 574 (15.4) | |
| **Patient category** | | | | |
| New | 2225 (62.2) | 1353(37.8) | 3578(95.8) | 0.247 |
| Transferred in | 12 (63.2) | 7(36.8) | 19(0.5) | |
| Treatment after loss to follow-up | 23(69.7) | 10(30.3) | 33(0.9) | |
| Treatment failure | 19(76.0) | 6(24.0) | 25(0.7) | |
| Relapse | 37(67.3) | 18(32.7) | 55(1.5) | |
| Other previously treated | 19(79.2) | 5(20.8) | 25(0.7) | |
| **Type of TB** | | | | |
| Pulmonary Positive | 1120(65.5) | 591(34.5) | 1712(45.8) | 0.006 |
| Pulmonary negative | 1147(60.4) | 753(39.6) | 1900(50.9) | |
| Extra pulmonary | 64(55.2) | 52(44.8) | 116(3.1) | |
| Not documented | 3(42.8) | 4(57.2) | 7(0.2) | |
| **HIV status** | | | | |
| Positive | 353(49.6) | 359(50.4) | 712(19.1) | <0.001 |
| Negative | 1619(66.6) | 813(33.4) | 2432(65.1) | |
| Not documented | 363(61.4) | 228(38.6) | 591(15.8) | |
| **District of enrolment** | | | | |
| Central Tongu | 120(60.9) | 77(39.1) | 197(5.2) | <0.001 |
| Hohoe | 300(72.8) | 112(27.2) | 412(11.0) | |
| Kadjebi | 224(68.3) | 104(31.7) | 328(8.8) | |
| Keta | 280(63.6) | 160(36.4) | 440(11.8) | |
| Ketu South | 661(58.8) | 464(41.2) | 1125(30.1) | |
| Kpando | 238(64.8) | 129(35.2) | 367(9.8) | |
| Krachi East | 100(52.9) | 89(47.1) | 189(5.1) | |
| Krachi West | 78(63.4) | 45(36.6) | 123(3.3) | |
| Nkwanta South | 209(60.6) | 136(39.4) | 345(9.2) | |
| South Tongu | 125(59.8) | 84(40.2) | 209(5.6) | |
| **Year of enrolment** | | | | |
| 2013 | 469(61.2) | 298(38.9) | 767(20.5) | 0.240 |
| 2014 | 481(60.6) | 313(39.4) | 794(21.3) | |
| 2015 | 476(65.8) | 247(34.2) | 723(19.4) | |
| 2016 | 432(63.3) | 251(36.8) | 683(18.3) | |
| 2017 | 477(62.1) | 291(37.9) | 768(20.6) | |

## HIV testing and prevalence of TB/HIV co-infection

Of the 3,735 TB patients enrolled on treatment during the period, 3144 (84.2%) were tested for HIV, of whom 712 (22.6%; 95%CI: 21.2–24.2) had positive HIV results documented. The TB/HIV co-infection rate was highest among children under 15 years of age (34.1%; 95%CI: 26.2–42.6) and lowest (8.6%; 95%CI: 6.3–11.5) among the elderly (>64 years). Males had a higher

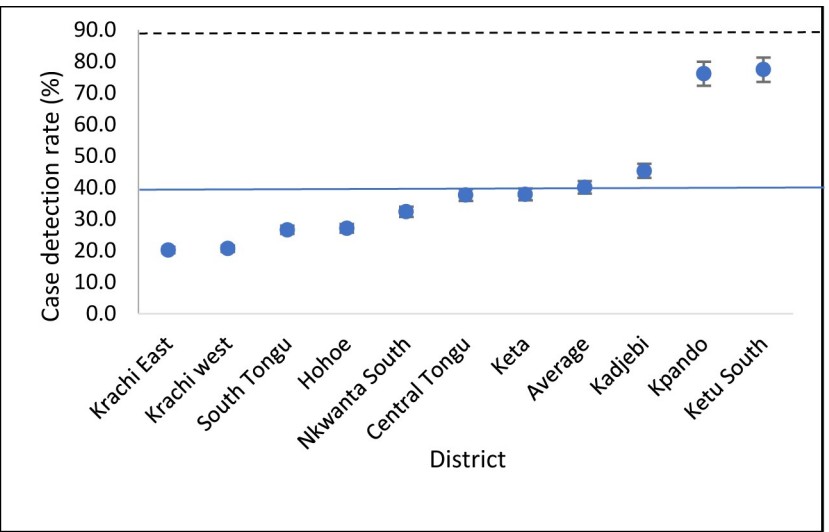

**Fig 1. TB CDR in Volta region, Ghana (2013–2017) by district.** The dotted line represent the end TB strategy target (>90%) and the blue line represent the study's overall CDR (40.1%).

prevalence rate (30.6%; 95%CI: 28.0–33.4) compared to females (17.9%; 95%CI: 16.3–19.7). Among the categories of TB patients, treatment after lost to follow-up patients had the highest co-infection rate (33.3%), followed by new cases (22.7%), and relapses (20.5%). The prevalence was 29.1% among smear-negative pulmonary TB patients compared to 6.6% for smear-positive pulmonary TB patients, while extra-pulmonary patients had 12% co-infection rate. Among the districts, the co-infection rate was highest in Ketu South (33%), Central Tongu (30.2%), and South Tongu (28%), all located in the southern part of Volta region; and lowest in Krachi East (9.2%) and Krachi West (11.9%) districts located in Northern Volta. With regards to the year of enrolment, the proportion of TB patients tested for HIV significantly increases from 72.5%

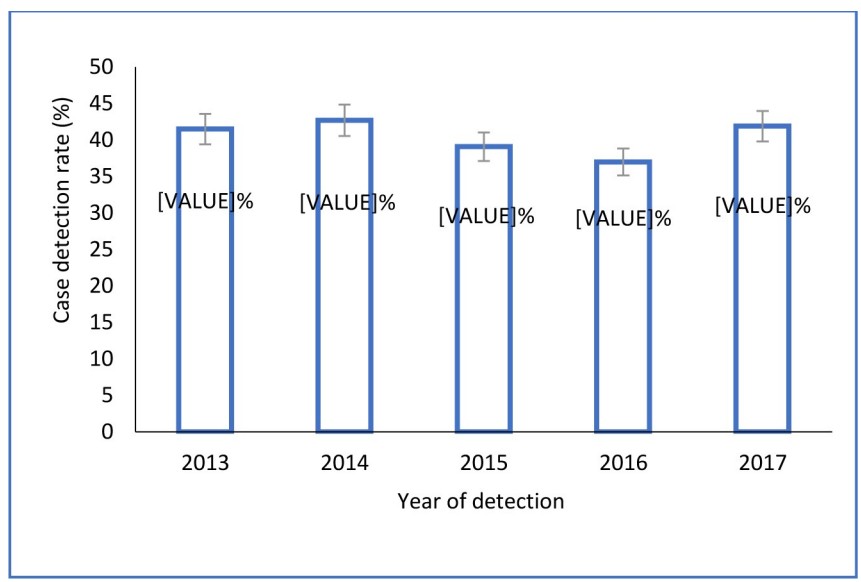

**Fig 2. Trends of TB CDR in 10 districts of Volta region, Ghana.**

in 2013 to 94.1% in 2017 (p<0.0001), however, the co-infection rate remained fairly above 20% in all the years (Table 3).

## TB case fatality rates and trends

Fig 3 shows the Overall trends in case fatality rates (CFR) stratified by HIV status. Of the 3,735 TB patients enrolled on treatment during the study period, 486 (13%; 95%CI: 12.0–14.1) of

**Table 3. HIV testing and TB/HIV co-infection rate for selected demographic and clinical characteristics.**

| Characteristics | #TB cases | TB cases tested for HIV(n%) | HIV Positive TB n (%) | 95%CI |
|---|---|---|---|---|
| **Age group(years)** | | | | |
| ≤14 | 161 | 138 (85.7) | 47 (34.1) | 26.2–42.6 |
| 15–24 | 311 | 261 (83.9) | 28 (10.7) | 7.2–15.1 |
| 25–34 | 601 | 509 (84.7) | 151 (29.7) | 25.7–33.8 |
| 35–44 | 852 | 728 (85.4) | 214 (29.4) | 26.1–32.8 |
| 45–54 | 743 | 617 (83.0) | 142 (23.0) | 19.7–26.5 |
| 55–64 | 493 | 415 (84.2) | 89 (21.4) | 17.6–25.7 |
| >64 | 574 | 476 (82.9) | 41 (8.6) | 6.3–11.5 |
| **Sex** | | | | |
| Male | 1400 | 1172 (83.7) | 359 (30.6) | 28.0–33.4 |
| Female | 2335 | 1972 (84.5) | 353 (17.9) | 16.3–19.7 |
| **Patient category** | | | | |
| New | 3578 | 3011 (84.2) | 684 (22.7) | 21.2–24.3 |
| Transferred in | 19 | 14 (73.7) | 2 (14.3) | 1.78–42.8 |
| Treatment after loss to follow-up | 33 | 30 (90.9) | 10 (33.3) | 17.3–52.8 |
| Treatment failure | 25 | 24 (96.0) | 3 (12.5) | 2.6–34.3 |
| Relapse | 55 | 49 (89.1) | 10 (20.5) | 10.2–34.3 |
| Other previously treated | 24 | 16 (66.7) | 3 (18.8) | 4.0–45.6 |
| **Type of TB** | | | | |
| Pulmonary Positive | 1712 | 1510 (88.2) | 251 (16.6) | 14.8–18.6 |
| Pulmonary negative | 1900 | 1528 (80.4) | 444 (29.1) | 26.8–31.4 |
| Extra pulmonary | 116 | 100 (86.2) | 12 (12.0) | 6.3–20.0 |
| Not documented | 7 | 6 (85.7) | 5 (83.3) | 35.8–99.6 |
| **Districts of enrolment** | | | | |
| Central Tongu | 197 | 149 (75.6) | 45 (30.2) | 22.9–38.3 |
| Hohoe | 412 | 400 (97.1) | 66 (16.5) | 12.9–20.5 |
| Kadjebi | 328 | 290 (88.4) | 51 (17.6) | 13.4–22.5 |
| Keta | 440 | 438 (99.5) | 66 (15.1) | 11.8–18.8 |
| Ketu South | 1125 | 884 (78.6) | 292 (33.0) | 29.9–36.2 |
| Kpando | 367 | 362 (98.6) | 86 (23.8) | 19.5–28.5 |
| Krachi East | 189 | 130 (68.8) | 12 (9.2) | 4.8–15.6 |
| Krachi West | 123 | 109 (88.6) | 13 (11.9) | 6.5–19.5 |
| Nkwanta South | 345 | 218 (63.2) | 35 (16.1) | 11.4–21.6 |
| South Tongu | 209 | 164 (78.5) | 46 (28.0) | 21.3–35.6 |
| **Year of enrolment** | | | | |
| 2013 | 767 | 556 (72.5) | 114 (20.5) | 17.2–24.1 |
| 2014 | 794 | 597 (75.2) | 140 (23.5) | 20.1–27.1 |
| 2015 | 723 | 630 (87.1) | 139 (22.1) | 18.9–25.5 |
| 2016 | 683 | 638 (93.4) | 136 (21.3) | 18.2–24.7 |
| 2017 | 768 | 723 (94.1) | 153 (21.2) | 18.2–24.3 |
| **Overall** | **3735** | **3144 (84.2)** | **712 (22.6)** | **21.2–24.2** |

them were reported dead. There were 260 (10.7%; 95%CI:9.5–12.0) TB deaths among HIV-negative people during the 5 years, and an additional 148 (20.8%; 95%CI: 18.0–23.9) deaths among HIV-positive patients (p<0.001). The burden of TB deaths varied enormously among districts (range: 5–21.6%), with the highest reported in Central Tongu, South Tongu, Hohoe, and Keta (21.6%, 95%CI: 16.2–27.6; 20.7%, 95%CI: 15.7–26.6; 15.3%, 95%CI: 10.8–17.5 and 15%, 95%CI: 12.0–18.6) respectively, while Kadjebi (9.6%, 95%CI: 6.7–13.1), Krachi West (5%, 95%CI: 2.3–10.2) and Krachi East (3.1%, 95%CI: 1.1–6.0) had the lowest mortality rate. In 2016, the overall CFR was 14.8%, up from 9.5% in 2012 (an increase of 35.8%) and from 13.4% in 2015. Among HIV-negative patients, the rate rose faster, from 7.4% in 2012 to 13.3% in 2016 (an increase of 44.4%), while death rate among HIV-positive patients increased by 19.8% during the same period. There was however no evidence of a difference in the rise in annual TB mortality in all cases (Fig 3). Generally, Hohoe, Ketu South, and Krachi East had increasing trends in CFR rates, while the rest of the districts showed irregular trends (Fig 4).

## Predictors of TB deaths

Table 4 shows the crude and adjusted logistic regression results of predictors of TB mortality. Multiple regression analysis of all forms of TB showed that infection with HIV, smear-positive pulmonary TB, and district of TB treatment were independently associated with TB deaths. HIV-infected patients had 2.10 odds of dying during TB treatment compared to HIV-negative patients (95%CI: 1.67–2.65; P<0.001). Similarly, smear-positive pulmonary TB patients were 0.66 times (95% CI: 0.54–0.82; p<0.001) less likely to die during anti-TB treatment compared to smear-negative pulmonary TB patients. Compared to Ketu South, TB patients who were enrolled on treatment at Central Tongu, Hohoe, Keta and South Tongu were more likely to die during treatment, while those treated in Krachi East had 29% (95%CI: 0.13–0.76; P<0.01) decreased odds of dying. Factors such as age, sex and category of TB patients did not predict TB deaths.

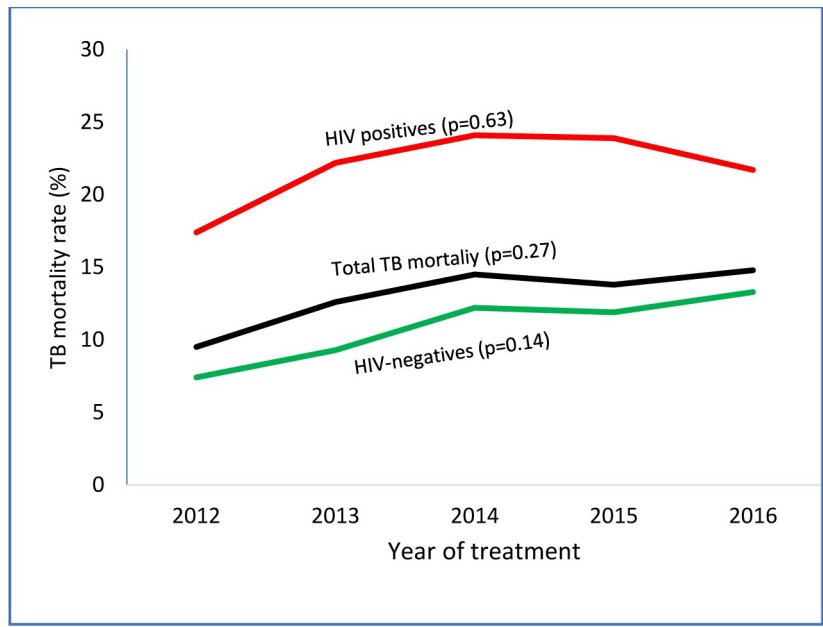

**Fig 3. Trends of TB case fatality rate stratify by HIV status in 10 districts of Volta region, Ghana, 2012–2016.** The mortality rates of HIV-positive TB are shown in **red**, rates of HIV-negative TB are shown in **green** and rates of all forms of TB are shown in **black**.

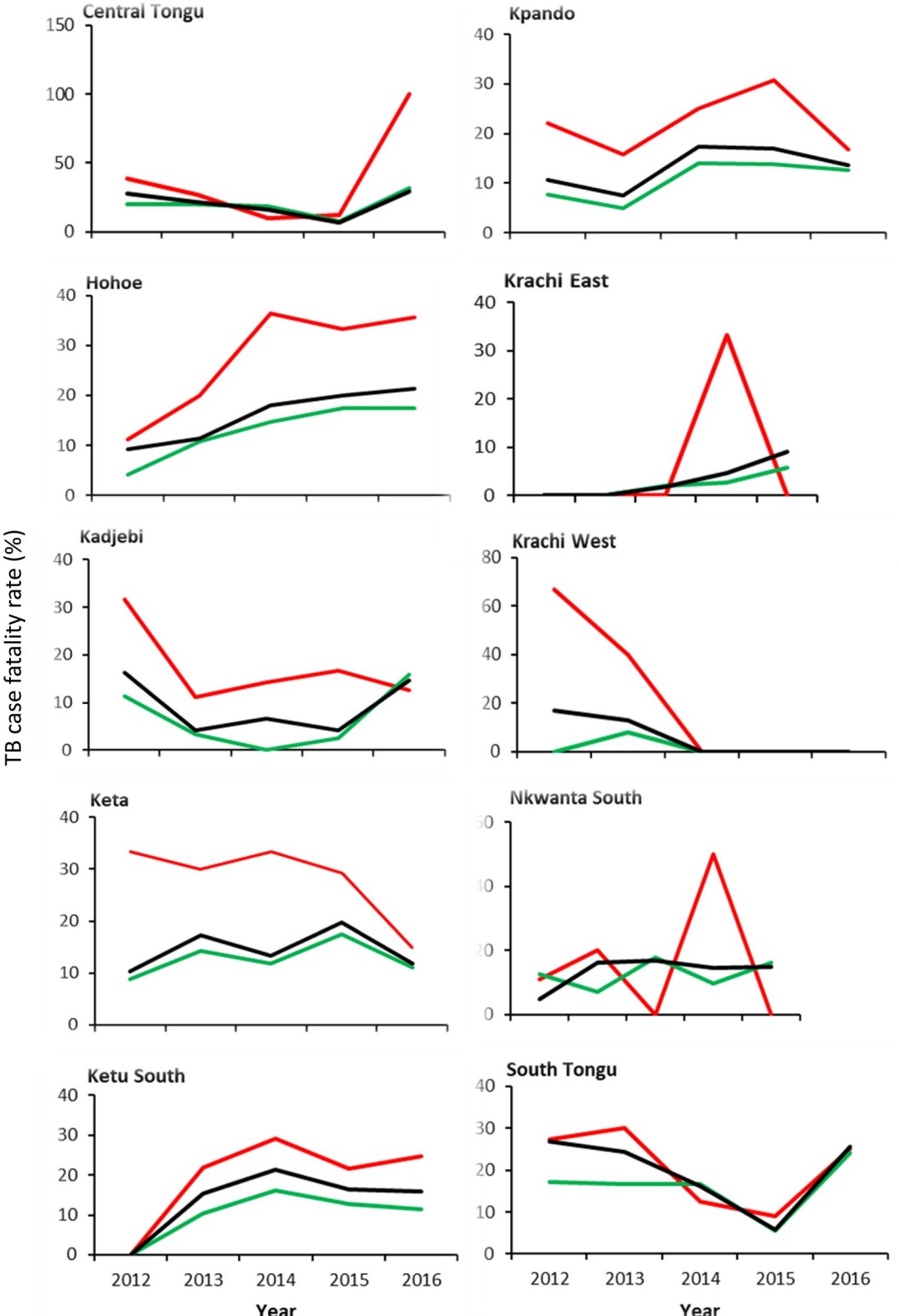

**Fig 4. District trends in TB case fatality rate in 10 districts of the Volta region, Ghana, 2012–2016.**

**Table 4. Logistic regression of predictors of TB mortality during treatment.**

| Variable | Died n (%) | COR | 95%CI | P-value | AOR* | 95%CI | P-value |
|---|---|---|---|---|---|---|---|
| **Age group(years)** | | | | | | | |
| <15 | 20(13.1) | 1 | | 0.413 | | | |
| 15–24 | 34(11.3) | 0.85 | 0.47–1.53 | | | | |
| 25–34 | 78(13.6) | 1.05 | 0.62–1.77 | | | | |
| 35–44 | 110(13.4) | 1.03 | 0.62–1.71 | | | | |
| 45–54 | 102(14.3) | 1.11 | 0.66–1.86 | | | | |
| 55–64 | 77(16.1) | 1.27 | 0.75–2.16 | | | | |
| >64 | 65(11.6) | 0.87 | 0.51–1.49 | | | | |
| **Sex** | | | | | | | |
| Male | 311(13.8) | 1 | | 0.467 | | | |
| Female | 175(12.9) | 0.93 | 0.76–1.13 | | | | |
| **Patient category** | | | | | | | |
| New | 466(13.5) | 1 | | 0.364 | | | |
| History of previous treatment | 20(16.4) | 1.26 | 0.77–2.05 | | | | |
| **HIV status** | | | | | | | |
| Negative | 260(10.7) | 1 | | | 1 | | |
| Positive | 148(20.8) | 2.25 | 1.80–2.82 | <0.001 | 2.10 | 1.67–2.65 | <0.001 |
| Unknown | 78(13.9) | 1.30 | 0.99–1.71 | | 1.28 | 0.96–1.72 | 0.095 |
| **Type of TB** | | | | | | | |
| Pulmonary negative | 290(15.7) | 1 | | <0.001 | 1 | | |
| Extrapulmonary | 9(8.4) | 0.49 | 0.24–0.98 | | 0.62 | 0.31–1.26 | 0.187 |
| Pulmonary positive | 185(11.3) | 0.68 | 0.56–0.82 | | 0.66 | 0.54–0.82 | <0.001 |
| **Districts of treatment** | | | | | | | |
| Ketu South | 140 (12.4) | 1 | | <0.001 | 1 | | |
| Central Tongu | 42 (21.6) | 1.84 | 1.25–2.71 | | 2.16 | 1.45–3.20 | <0.001 |
| Hohoe | 57 (15.3) | 1.21 | 0.87–1.69 | | 1.64 | 1.14–2.33 | 0.007 |
| Kadjebi | 31 (9.6) | 0.71 | 0.47–1.07 | | 0.87 | 0.58–1.32 | 0.529 |
| Keta | 66 (15.0) | 1.18 | 0.86–1.63 | | 1.53 | 1.09–2.13 | 0.012 |
| Kpando | 48 (13.2) | 1.02 | 0.72–1.46 | | 1.19 | 0.83–1.71 | 0.345 |
| Krachi East | 5(3.1) | 0.21 | 0.08–0.53 | | 0.29 | 0.12–0.72 | 0.008 |
| Krachi West | 6 (5.0) | 0.36 | 0.15–0.83 | | 0.48 | 0.21–1.14 | 0.100 |
| Nkwanta South | 48 (14.3) | 1.12 | 1.79–1.59 | | 1.35 | 0.94–1.95 | 0.101 |
| South Tongu | 43 (20.7) | 1.76 | 1.20–2.57 | | 2.12 | 1.44–3.15 | <0.001 |

*adjusted for HIV status, type of TB and district of treatment

## Discussion

### TB case detection rate and trends

Using retrospective surveillance data, we assessed 5-year trends of TB case detection, mortality and co-infection with HIV and compared the rates among 10 districts in the Volta region of Ghana. The study further assessed patients demographic and disease characteristics as independent predictors of TB deaths. Our analysis underscores significant district variation in the CDR and that underreporting of TB is a major problem in this part of the country, with an average of only 4 in 10 of the estimated cases being reported to the NTBCP. This translates to over 5,500 cases who remained undiagnosed or likely to have been given treatment by providers who are not connected to the NTBCP. Our finding falls short of the 90% target prescribed

by the WHO's end TB strategy for all national control programmes in order to reduce the number of people falling ill with TB by 90% and TB deaths by 95% by 2035 [4]. Availability and proximity of diagnostic facilities and DOTS centres are likely to be an essential factor for case detection since districts with poor CDR such as Krachi East and Krachi West had the lowest diagnostic facility density, whereas Kpando and Ketu South with the highest laboratory and DOTS centre density [13] had the highest CDR above the 70%. This was also found by Obasanya and others [14] in their Nigerian study. The CDR appeared to have stagnated during the study period in all districts studied, a situation that has been partly attributed to the insufficient motivation of health personnel to detect and manage cases, dwindling of funds, inadequate private sector involvement, inadequate diagnostic centers and poor health-seeking behaviour of patients [15,16]. The most recent (2014) Ghana Demographic and Health Survey, for instance, reported that knowledge regarding TB was high among the populace. However, about 33% of women and 25% of men in the country would not disclose their TB disease status to their families, which may be driven by stigma [17]. The evidence for the best approach to increase TB case detection is incomplete [18], however, active case-finding has been identified as one such approach that has been successful [19]. A study in Vietnam, for instance, has highlighted the value of household contact tracing and investigation [20,21]. Other countries, such as India focuses on systematic screening in high-risk populations such as people living in urban slums [22]. Ohene and colleagues [23] however argue that, when considering a TB screening program, it is essential to simultaneously look at the overall health system functions and enhance capacity to facilitate early detection. Patient delay for seeking TB care may arise from various factors, including barriers in access to care such as costs of seeking care or stigma [24]. Addressing these barriers might have the secondary but important effect of encouraging symptomatic individuals to seek care.

## HIV test among TB patients and TB/HIV co-infection rate

The deadly synergy between TB and HIV raises complexity and represent a considerable obstacle towards achieving global TB elimination target [25]. The offer of HIV test for TB patients presents an opportunity essential for providing the best care for TB/HIV coinfected patients, including antiretroviral therapy [2]. In this present study, an average of 84% of TB patients notified during the period had documented HIV test result. This is lower than the national average of more than 91% in 2017 [2] and what Osei and colleagues [26] previously reported in their study also in the Volta region of Ghana (92%). The percentage of TB patients with documented HIV results however, increased substantially from nearly 73% to more than 94% from 2013 to 2017 (a 23% rise), consistent with regional [13] and the global trends [2], an indication that the world is on track to achieve the 100% HIV testing coverage among TB patients by the 2025 target set in the End TB Strategy. The rate of documented HIV status among TB patients, however, varied considerably among districts studied, with the highest rate of over 95% achieved in Hohoe, Keta and Kpando districts and the lowest of below 70% found in Krachi East, and Nkwanta South districts. The reason for this variation could be attributed to the availability of trained personnel at DOTS centres in the districts to offer routine HIV counselling and testing.

Regional and countries prevalence studies have reported varied magnitude of TB/HIV co-infection rate, ranging between 2.9 and 72.3%, with an average rate of 23.5% [27]. Our pooled analysis from 10 districts in the Volta region of Ghana from 2013 to 2017 found a high co-infection rate among the cohort of TB patients. The 22.6% TB/HIV coinfection rate estimated in this study is higher than the national average of 19% estimated by the WHO in 2018 [7], what was reported in a previous study in the same region by Osei and colleagues (18.2%) [26],

and that reported in a recent national survey (14.7%) [28]. The rate is however lower than an estimate from the WHO's African region (29%) and comparable to findings from other studies done in Ethiopia [29,30]. According to the sub-group analysis, the HIV infection rate among TB patients was highest among children under 15 years of age (30.1%) followed by people between 25 and 44 years. This finding is consistent with previous finding from the same study region [26] and that of other reports that have shown higher rate in persons aged 26–45 years who represent the sexually active age group [31,32]. Contrary to the previous report [26] however, males had the highest TB/HIV coinfection rate (30.6%) compared to 17.9% among females in this study. Furthermore, Ketu South, Central Tongu, South Tongu and Kpando, all urban districts had TB/HIV rate higher than the study average, while Krachi East and Krachi West (rural districts) recorded the lowest co-infection rate. The difference between the districts might be due to the difference in the demographic characteristics of the population and associated HIV/AIDS prevalence. The 2014 Ghana Demographic and Health Survey, for instance, reported that HIV/AIDS is more prevalent in urban areas than rural areas [17]. This sounds logical since HIV infection is known to be associated with an increased risk of developing active TB by accelerating disease progression [8,9]. Overall, the rate of TB/HIV co-infection has been falling globally since 2008 [2]. In this present study, however, the rate has stagnated around just over 20%, which is not different from what was previously reported from the same region [26]. This implies that activities to reduce HIV-associated TB need to be strengthened, which may include prevention, early detection and effective case management.

## Trends and predictors of TB mortality

Tuberculosis case fatality rate (CFR) is an important indicator for monitoring progress towards the 2020 and 2025 milestones. In order to achieve the 2025 global milestone for reductions in TB deaths, a CFR of 6% or less is required by all NTPs [2]. The 13% overall CFR estimated in this present study compares favourably with what a previous South African study found (16.3%) in a 10-year electronic record review [33] and in Zimbabwe (22%) [34] but higher than findings reported in Ethiopia [35,36], Cape Town (3.7%) [37], Washington state, US (12.1%) [38] and Taiwan (12.3%) [39]. The differences in CFR in different settings might be due to the differences in the study sites, study period, the study design, and target population. Differences in the quality of case management cannot also be ruled out. Our results demonstrate that the burden of TB deaths vary considerably among districts, ranging from 5–21.6%, with the Southern (Central Tongu, South Tongu and Keta) and the Middle (Hohoe) belts recording the highest CFR, whereas districts in the Northern belt (Krachi East, Krachi West and Kadjebi) having the lowest mortality rates, suggesting relatively access to better case management in Northern Volta.

Globally, TB mortality rate fell by 42% between 2000 and 2018 (including 3.6% between 2017 and 2018) [7]. Conversely, our analysis shows considerable (35.8%) rise in the mortality rate between 2012 and 2016 (including 10% between 2015 and 2016). Additionally, unlike the global trend, the mortality rate for both HIV-negative and HIV-positive patients rose by 44.4% and 19.8% respectively. This suggests that the region is not on track to reach the End TB Strategy milestone of a 35% reduction in the total number of TB deaths between 2015 and 2020 since reaching the 2020 milestone requires a CFR of no more than 6% by 2020 [7].

Consistent with many other studies [26,40], we found that HIV-positive TB patients had increased odds of dying from the disease. This could potentially be attributed to drug interactions between the rifamycins (rifampin or rifabutin) and some antiretroviral agents [41], which may occur when obliviously given together, as well as malabsorption of anti-TB drugs among patients with advanced HIV [42], leading to low serum concentrations of drugs which

may lead to adverse treatment outcome [43]. Among HIV-infected individuals, cotrimoxazole prophylaxis therapy (CPT) has been known to reduce mortality by about 25–46% in Sub-Saharan Africa even in areas with high antimicrobial resistance. [44,45]. Studies from different populations [35,46,47] have also shown that TB/HIV co-infected patients who were not taking CPT were at highest risk of death. This study, however, did not have the data to confirm or refute this claim. Additionally, we found that patients who were pulmonary smear-positive were 0.66 times less likely to die during anti-TB treatment, compared to smear-negative pulmonary TB patients. Our finding corroborates with what Alobu et al. also found in their Nigerian study, where TB patient with negative sputum smear had 2.4 higher odds of dying during treatment compared to sputum smear positive patients [46]. This could be due to delays in the establishment of diagnosis among smear negative pulmonary TB patients [48]. Clinicians often are reluctant to treat TB patients whose diagnosis are not certain as a result of strict rules imposed on them by public health policies [49], which may result in further delays in initiation of anti-TB therapy that can lead to worse disease prognosis. Additionally, the presence of advanced HIV infection has been linked to atypical clinical presentation of TB, including smear-negative pulmonary TB and normal chest X-ray, making it difficult to diagnose TB [50]. In a systematic review and meta-analysis to identify the predictors on in-hospital deaths among pulmonary TB patients however, Carlos and colleagues did not find an association between positive sputum smear and in-hospital mortality [51].

Using Ketu South as a reference district, patients who had their anti-TB treatment from Central Tongu, Hohoe, Keta and South Tongu had increased odds of dying during treatment, whereas those who were treated at Krachi East were less likely to die. This variation may be due to differences in the quality of case management as well as differences in TB/HIV co-infection rate.

Our study is not without limitations. Firstly, due to the retrospective nature of the study, we could not independently corroborate the accuracy of data used in the analysis, nor could we collect additional data needed to confirm or refute our findings. Secondly, the presence of residual confounders is possible since data were either incomplete or not available to gather the necessary variables such as patient's compliance, nutritional status, and the use of ART, CPT and other prophylaxis. Lastly, it is possible that some patients hid their history of previous anti-TB treatment at the time of diagnosis and were misclassified under a different category. Despite these limitations, the study did provide useful information on the trends of TB case detection rate, co-infection with HIV, and predictors of mortality in ten districts of the Volta region. Findings may be useful for TB programme planning and implementation.

## Conclusion

Our analysis found low and stable TB case detection rate in the Volta Region of Ghana, with substantial district variations. Also, the prevalence of TB-HIV co-infection was quite high, more prevalent in urban districts, and remained steady over the years. The study further found high and increasing trends of TB case fatality rate, which was influenced by HIV infection, sputum smear-negative and treatment setting. Our results underscore three key needs for improving TB control in the region: 1) the need to adopt practical and feasible approaches for active case finding, which may involve training of local health personnel and volunteers, strengthening public-private partnerships and increasing laboratory density, in order to increase case detection. 2) embark on primary prevention programmes to reduce HIV infection among the general populace, and 3) improve case management, particularly among patients co-infected with HIV through early detection and initiation of ART, CPT or other prophylaxis in order to mitigate the burden of co-morbidity and to reduce death.

## Supporting information

**S1 File. Anonymized data set used for this study.**
(XLSX)

## Acknowledgments

Authors are grateful to health authorities and staff, particularly the TB coordinators of all participating districts for their support and permission to carry out and publish this study. We acknowledge the following persons who collected data; Daniel Adanfo, Bless Ativor Doepe, Andrews Owusu, and Augustine Goma Kupour, all past students of School of Public Health, University of Health and Allied Sciences, Ho, Ghana.

## Author Contributions

**Conceptualization:** Eric Osei.

**Formal analysis:** Eric Osei, Samuel Oppong.

**Investigation:** Eric Osei, Samuel Oppong.

**Methodology:** Eric Osei.

**Project administration:** Eric Osei.

**Supervision:** Joyce Der.

**Writing – original draft:** Eric Osei, Joyce Der.

**Writing – review & editing:** Eric Osei, Samuel Oppong, Joyce Der.

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
