## [Decision Letter · Decision Letter 0]

27 Apr 2020

PONE-D-20-02837

Trends of tuberculosis case detection, mortality and co-infection with HIV: lessons from and for the National Control Programme

PLOS ONE

Dear Mr osei,

Thank you for submitting your manuscript to PLOS ONE. After careful consideration, we feel that it has merit but does not fully meet PLOS ONE’s publication criteria as it currently stands. Therefore, we invite you to submit a revised version of the manuscript that addresses the points raised during the review process.

We would appreciate receiving your revised manuscript by Jun 11 2020 11:59PM. To enhance the reproducibility of your results, we recommend that if applicable you deposit your laboratory protocols in protocols.io, where a protocol can be assigned its own identifier (DOI) such that it can be cited independently in the future. For instructions see: http://journals.plos.org/plosone/s/submission-guidelines#loc-laboratory-protocols

We look forward to receiving your revised manuscript.

Kind regards,

Mark Spigelman

Academic Editor

PLOS ONE

Additional Editor Comments:

Dear Dr Francis Mhimbira

Your article has been reviewed and whilst we believe it has merit it does need some significant changes before we would consider it for possible publication.

Please look at the comments made by the two referees and make the suggested changes and then submit once again for consideration.

M Spigelman

2. Please refer to any post-hoc corrections to correct for multiple comparisons during your statistical analyses. If these were not performed please justify the reasons.

Please refer to our statistical reporting guidelines for assistance (https://journals.plos.org/plosone/s/submission-guidelines.#loc-statistical-reporting).

5. Your ethics statement must appear in the Methods section of your manuscript. If your ethics statement is written in any section besides the Methods, please move it to the Methods section and delete it from any other section. Please also ensure that your ethics statement is included in your manuscript, as the ethics section of your online submission will not be published alongside your manuscript.

Reviewers' comments:

Reviewer's Responses to Questions

**Comments to the Author**

1. Is the manuscript technically sound, and do the data support the conclusions?

Reviewer #1: Yes

Reviewer #2: Partly

2. Has the statistical analysis been performed appropriately and rigorously? 

Reviewer #1: I Don't Know

Reviewer #2: No

3. Have the authors made all data underlying the findings in their manuscript fully available?

Reviewer #1: No

Reviewer #2: Yes

4. Is the manuscript presented in an intelligible fashion and written in standard English?

Reviewer #1: Yes

Reviewer #2: Yes

5. Review Comments to the Author

Reviewer #1: This paper explores the trends in TB case detection rate and mortality across randomly selected districts in Ghana. There is some descriptive analysis as well as logistic regression of factors contributing to death.

Major points

- I was concerned about the link between smear positive disease and lowered death rate vs. smear negative disease. It is mentioned in the discussion with some reasonable explanations but could you also add in whether this has been seen in other settings?

Minor points

- you mention in the introduction that no one has looked at trends in CDR / mortality in Ghana - what about the WHO? Perhaps no one has looked at the lower district level that you are able to?

- why did you use a different time period for treatment vs. mortality rate predictions? what impact might this have on your results?

- how often was the data extraction systematically checked?

-what is the p-value in Table 2 - univariate?

- table 3 - how was the 95% CI generated?

- bottom of the paragraph entitled: TB case fatality rates and trends, it should be "showed irregular trends" not "shown irregular trends".

- Table 5 - is this the multivariate analysis? did you just combine all factors into one regression?

- Discussion: I'm not sure that you want to say "confirmed by Obasanya", rather that this was "also found to be the case" in Nigeria.

- Discussion: replace "antibiotic microbial" resistance with just "antibiotic" resistance

- Figures: expand CDR and in general the descriptions of what is shown

- Figure 1: label the target - Ghana's / WHO's / Global TB?

- Figure 1: what is the blue line?

- Figure 2: what are the numbers? mean?

- Figure 3: please take the descriptions of what the colours mean from FIgure 4 and add to Figure 3

- Figure 4: what happened in Ketu South in 2012?

Reviewer #2: Trends of tuberculosis case detection, mortality and co-infection with HIV: lessons from and for the National Control Programme.

General comments

The authors have written an interesting article on trends of TB notification, mortality and TB/HIV co-infection. They have explored well the reasons for observing such trends and suggested areas to improve TB notification, reduce mortality and HIV co-infection in the study (efforts can apply to the whole country). The article is well written, but there are few comments that need to be addressed to bring the paper to the quality worth of publishing.

General comments to be considered once they revise the manuscript:

• Add continuous line numbers for easy commenting.

• Corresponding author name: why the name is all in small letters (abstract)?

• Where applicable, use non-stigmatizing language like cases to TB patients.

ABSTRACT

Background:

• ….” across a geographical location and to provide…” consider replacing the comma with and.

Methods:

• Bivariate and multivariate logistic regression were used to identify predictors of TB deaths with Odds ratios and 95% confidence intervals estimated. What is the difference between bivariate and multivariate logistic regression? I would suggest you use simple and multiple logistic regression term.

Results:

• no results of logistic regression analysis.

Conclusion

• how is high co-infection with HIV a proxy for week programme performance?

• TB case detection rate was low and remained stable, whereas co-infection with HIV and mortality rates were quite high, suggesting weak programme performance. Please consider revising the conclusion based on the findings. Comment on technical aspects of why TB notification is low and provide a normative guidance of how to improve TB notification.

MAIN ARTICLE

Background

• Worldwide, about 10 million incident cases of TB occur annually. Though correct, please consider mention the 10 million estimate corresponds to which year.

• Include estimates of death at global and WHO AFRO region, as mortality is one of the outcomes of interest.

Methods

• Categorical variables were expressed in percentages and mean ± standard deviation (SD) for continuous variables, they were compared using Pearson’s chi-square test or Fisher’s exact test and the independent-samples t-test accordingly to describe the study population. The presentation of continuous variables into mean or median is dependent on their distribution. How did the authors make a decision to present directly means and SD? In your results, first paragraph, you have presented median and IQR.

• Inclusion of variables in the regression model could be due to biological plausibility in their association and not necessary for variables with p-values of <0.05.

• Case Detection Rate (CDR) was computed as the number of TB cases notified during the period divided by the estimated annual TB incidence per 100 000 Ghanaian population. Also, the reference is number 11 which is (World bank. The world bank data catalogue. http://datacatalog.worldbank.org/publiclicenses#cc-by), not sure if CDR definition and CDR data comes from world bank instead of WHO. The current term now used is the treatment coverage, do the authors care to explain their choice of using CDR instead of treatment coverage.

• Data sources and extraction: change information to data and what are the TB/HIV collaborative activities data? TB/HIV collaborative activities do you limit yourselves to HIV testing and ART initiation? Or you also include IPT? Please be clear.

• Case definition: for pulmonary positive TB patients, are does the definition also include those with GeneXpert results?

Results

• General results: presented median & IQR, but the not described this in methods section. The method section has only means and SD.

• How did the author calculate estimated TB incidence for each district to come up with the CDR? Need to be clear in the methods section.

• Table 5: need to add a note, that the adjusted p-value is adjusted to what variables.

• Logistic regression analysis: have the authors considered the effect of inter-cluster correlations? Can the author justify removing sex in the regression analysis?

• From results: Similarly, smear-positive pulmonary TB patients were 66% (95% CI: 0.54-0.82; p<0.001) less likely to die during anti-TB treatment compared to smear-negative pulmonary TB patients. If the point estimates is in %, then the 95% CI should also be in %.

Discussions

• Our analysis underscores significant district variation in the CDR and that underreporting of TB is a major problem in this part of the country, with an average of only 4 in 10 of the estimated cases being reported to the NTBCP. This statement is partly true, as under-reporting is one of the causes of such a discrepancy between incident and notified TB patients. However, under-diagnosis could also contribute to the gap. Are the authors confident that diagnostic capacity is optimal, and that the patients are diagnosed but not only reported. is this statement supported by an inventory study that documents under-reporting at least in those 10 districts?

• Our finding falls short of the 70% target prescribed by the WHO for all national control programmes to reduce the number of people falling ill with TB by 90% and TB deaths by 95% by 2035. I am interested on how authors came to the conclusion that the country country falls short of reducing TB incidence and mortality by 2035. I don’t see any projections to back up their conclusion in this one.

• The deadly synergy between TB and HIV raises complexity and represent a considerable obstacle towards achieving global TB elimination target. The WHO, therefore, recommends systematic screening and documentation of HIV status among all TB patients enrolled on anti-TB treatment as an essential component of the TB care package [24], which is one of the core indicators for monitoring the implementation of the End TB strategy at all levels. The offer of HIV test for TB patients presents an opportunity essential for providing the best care for TB/HIV coinfected patients, including antiretroviral therapy [2]. Please consider deleting or shorten to a sentence or two. Most of the text is in the introduction part.

• Tuberculosis Case Fertility rate (CFR). Please change to fatality.

• From results: Similarly, smear-positive pulmonary TB patients were 66% (95% CI: 0.54-0.82; p<0.001) less likely to die during anti-TB treatment compared to smear-negative pulmonary TB patients. From discussion section: Additionally, we found that patients who were pulmonary smear-positive were 34% less likely to die during anti-TB treatment, compared to smear-negative pulmonary TB patients. I am bit confused with the presentation of these findings.

6. PLOS authors have the option to publish the peer review history of their article (what does this mean?). If published, this will include your full peer review and any attached files.

Reviewer #1: No

Reviewer #2: No

---

## [Author Response · Author response to Decision Letter 0]

6 May 2020

response letter has been attached to the submission

---

## [Editor Report · Decision Letter 1]

4 Jun 2020

Trends of tuberculosis case detection, mortality and co-infection with HIV in Ghana: a retrospective cohort study

PONE-D-20-02837R1

Dear Dr. Eric Osei

We’re pleased to inform you that your manuscript has been judged scientifically suitable for publication and will be formally accepted for publication once it meets all outstanding technical requirements.

Kind regards,

Mark Spigelman

Academic Editor

PLOS ONE

Additional Editor Comments (optional):

Dear Dr Eric Osei

Thank you for the comprehensive review of your paper following suggestions made by your two referees.

I am satified that you have addressed the major issues raised and as such am recommending your paper as suitable for publication

M Spigelman
---

## [Editor Report · Acceptance letter]

15 Jun 2020

PONE-D-20-02837R1 

Trends of tuberculosis case detection, mortality and co-infection with HIV in Ghana: a retrospective cohort study 

Dear Dr. Osei:

I'm pleased to inform you that your manuscript has been deemed suitable for publication in PLOS ONE. Congratulations! Your manuscript is now with our production department. 

Kind regards, 

on behalf of

Dr. Mark Spigelman 

Academic Editor

PLOS ONE